# Is Obesity a Risk Factor for Periodontal Disease in Adults? A Systematic Review

**DOI:** 10.3390/ijerph191912684

**Published:** 2022-10-04

**Authors:** Ghadah Abu-Shawish, Joseph Betsy, Sukumaran Anil

**Affiliations:** 1Department of Dentistry, Oral Health Institute, Hamad Medical Corporation, Qatar University, Doha 3050, Qatar; 2Department of Periodontics, Saveetha Dental College and Hospital, Saveetha Institute of Medical and Technical Sciences, Chennai 600077, India; 3Pushpagiri Research Centre, Pushpagiri Institute of Medical Sciences and Research Centre Thiruvalla, Pathanamthitta 689101, India

**Keywords:** clinical attachment level, inflammation, obesity, periodontal diseases, periodontitis

## Abstract

There is inconclusive evidence about the link between the severity and prevalence of periodontitis in obese adults. Therefore, this systematic review aims to explore the possibility of significant evidence on the association between obesity and periodontitis and to determine the necessity to consider obesity as a risk factor for periodontitis. We followed the PRISMA protocol, and studies that met the eligibility criteria were included in this review. The risk of bias in individual studies was also evaluated. This review included 15 observational studies (9 cross-sectional studies, 2 case-control, and 4 cohort studies). The total study subjects from these studies were 6603 (males = 3432; females = 3171). Most studies showed a significant association between obesity and periodontitis. Among these studies, a few showed obese females to be at a higher risk, and one study found no association between obesity and periodontal disease at all. Based on the evidence obtained from this review, the body mass index (BMI) should be routinely assessed in patients to assess the risk for periodontal disease and to offer personalized management of periodontitis. Based on the findings of this review, we recommend the need to initiate awareness among clinicians and implement dental hygiene care prevention measures for obese patients.

## 1. Introduction

Obesity is a complex multifactorial chronic disease resulting in excessive fat deposition in the adipose tissue [1]. Obesity has been considered a common metabolic and nutritional disorder. It has emerged as a significant public health concern globally [2]. According to a recent WHO factsheet, the worldwide prevalence of obesity in the adult population has tripled over the last few decades and was recorded to be 13% in 2016. Obesity is a significant risk factor for various common health conditions such as diabetes, coronary disease, hypertension, osteoarthritis, and periodontitis [3,4].

In periodontitis, increased inflammation is due to a shift in the subgingival microbial ecology, which leads to continuous host-mediated destruction of the periodontium [5,6]. Studies have confirmed that obesity contributes to the severity of periodontal disease [5,7,8,9,10]. Many studies demonstrated that obesity increases the systemic inflammatory load due to increased metabolic and immune parameters, increasing susceptibility to periodontal disease [11,12]. A recent meta-review suggested that obesity could be a risk factor for the onset and progression of periodontal disease [13]. Another recent comprehensive review has found a possible link between periodontitis, metabolic syndrome, and obesity [14] based on plausible mechanisms showing how these conditions adversely affect each other.

In obesity, there is increased body fat deposition compared to lean body mass and may cause chronic, low-grade inflammation [15,16,17] with few or no symptoms [17,18]. Earlier, the role of adipose tissue was poorly understood as only an organ to store triglycerides. Extensive work in this field has shown that adipose tissue is an active endocrine organ with many metabolic functions. For example, adipose tissue releases many immunomodulatory factors that significantly regulate metabolic and vascular biology [19]. Adipose tissues release adipokines that exert their effect by acting either locally or systemically [20,21]. These adipokines have various actions, such as hormone-like proteins, pro-inflammatory cytokines, vascular hemostasis, and blood pressure regulation [22,23]. Thus, there could be a biological plausibility in the positive association between obesity and periodontitis.

Body mass index (BMI) is the most common method to measure obesity. WHO defines an adult as overweight if the BMI is greater than or equal to 25; and obese if BMI is greater than or equal to 30. BMI is a useful method to identify obesity as its calculation is the same for males and females and all adult age groups. It indicates overall adiposity and the lowest overall risk to health due to increased weight [24]. BMI is calculated by dividing weight in kilograms (kg) by square height in meters (m) [25] and categorized into four categories- underweight (less than 18.5 kg/m^2^), normal (18.5 to 24.9 kg/m^2^), overweight (25.0 to 29.9 kg/m^2^), and obese (more than 30.0 kg/m^2^). Other measures of obesity include waist circumference (WC), waist-to-hip ratio, and total body fat.

Several studies report a link between periodontal disease and systemic diseases. For example, in pregnant women, vitamin D deficiency and poor oral health resulted in a higher number of pre-term births and low birth weight [26]. Similarly, an association between periodontitis and cardiovascular diseases can be seen in the literature due to increased endothelial dysfunction and other arterial factors [27]. Epidemiologic studies also demonstrated a positive association between periodontal disease and cancer, especially for anatomic sites near the oral cavity [28]. Another link is between poor oral health and worse functional status in amyotrophic lateral sclerosis [29]. There are several existing studies on the association between periodontitis and obesity. Still, there is no conclusive evidence about the link between the severity and prevalence of periodontitis in obese adults. Therefore, this systematic review aims to explore the possibility of significant evidence on the association between obesity and periodontitis and to determine the necessity to consider obesity as a risk factor for periodontitis.

## 2. Materials and Methods

The research question for this systematic review was—Is periodontitis (in terms of clinical attachment level (CAL)) more prevalent and severe in adults with increased BMI?

Is periodontitis measured by CAL more prevalent and severe in patients with a BMI of more than 30?Does BMI more than 30 affect tooth loss due to periodontitis and bleeding on probing?

### 2.1. Protocol and Registration

The research problem, focus question, and criteria for inclusion and exclusion were identified after a preliminary literature study. Following this, a protocol was prepared and registered with PROSPERO before the commencement of the review (CRD42022340119).

### 2.2. Eligibility Criteria

The PECO (Population, Exposure, Comparisons, Outcomes) framework was used to formulate the focus question: Among patients between the ages of 18–70, is periodontitis (in terms of CAL) more prevalent and severe in adults with increased BMI? Does obesity increase tooth loss and bleeding on probing? Population (P) = Patients between 18–70 years of age; Exposure (E) = high BMI/Obesity; Comparison (C) = normal BMI; Outcome (O) = periodontal attachment loss (CAL and PPD).

Studies with the following criteria were included (1) patients between 18–70 years of age; (2) males and females, (3) patients diagnosed with overweight and obesity based on WHO criterion (BMI above 25 and 30 respectively), (4) patients diagnosed with periodontitis based on CAL and probing pocket depth (PPD) (5) case-control studies, cross-sectional studies, as well as cohort studies.

The following criteria were used to exclude studies- (1) patients younger than 18 years and older than 70 years, (2) studies that did not directly evaluate the relation between periodontal disease and obesity, (3) studies that include confounding factors such as diabetes, smoking, poor oral health, hormonal changes such as pregnancy and menopause. Since we found only two studies without any confounding factors, we also included those studies that reported diabetes and smoking) (4) studies that describe systemic diseases in obese patients (5) studies in which no BMI estimation is given (6) studies not evaluating periodontitis using CAL as an outcome. Moreover, case reports, abstracts, editorials, letters, comments to the editor, reviews, meta-analyses, book chapters, and articles not written in English were excluded.

### 2.3. Information Sources and Search Strategies

We followed the PRISMA protocol to conduct and report this systematic review (PRISMA (Preferred Reporting Items for Systematic Reviews and Meta-Analyses) statement [30]. Studies from 1 Jan 2000 to date (30 June 2022) that met the PECO criteria were included in this review. The search was carried out on all major databases, such as PubMed, Scopus, Web of Science Core Collection, and Cochrane, to identify observational studies on obesity and periodontal disease association.

The search was conducted based on the research question’s main two concepts (Periodontal Disease and Obesity). The literature database was searched using MeSH terms, keywords, and other free terms related to Periodontal Disease (“Periodontal disease” OR “periodontitis” OR “alveolar bone loss” OR, “Clinical Attachment loss” OR “Periodontal pocket” OR “CAL” OR “tooth loss” OR “Periodontal Attachment Loss” OR Periodont*) and Obesity (“Obesity” OR “obese” OR “body mass index” for identifying relevant publication up to 30 June 2022.

In addition, references to relevant studies and manual searching also were done for other potentially appropriate publications.

### 2.4. Studies Selection

Initially, the articles were identified through the database with the help of keywords. After this, relevant articles were screened with the help of the title and abstract by two reviewers (G.A. and B.J.) based on the inclusion and exclusion criteria. Subsequently, duplicate articles were removed with the help of a citation/reference manager (Endnote). Full texts of the remaining articles were retrieved and examined by two reviewers (G.A. and B.J.). A third reviewer (S.A.) was involved in resolving through discussion, and a final consensus was reached in case of any disagreement.

### 2.5. Data Extraction and Data Items

Piloting the study selection process was done to refine inclusion criteria. This also ascertained that the criteria could be applied consistently by more than one person. Data were retrieved from the selected studies and tabulated using excel sheets.

Two reviewers independently extracted the following data from cross-sectional, case-control, and cohort studies-: Author, year, country, study design, age range, sample size, male/female, study population, obesity criteria, the definition of periodontitis used, secondary parameters such as tooth loss and bleeding on probing, and main observation. Similar to study selection, the data was obtained and confirmed by two reviewers (G.A. and B.J.), and any disagreement was resolved through discussion with the help of a third reviewer (S.A.).

### 2.6. Diagnosis of Obesity

According to the WHO criteria, overweight and obesity are defined as abnormal or excessive fat accumulation that presents a health risk. A BMI over 25 is considered overweight, and over 30 is obese. For the present study, obese patients with a BMI above 30 were considered.

### 2.7. Diagnosis of Periodontitis

Patients diagnosed with periodontitis when at least CAL and PPD were included in this study. This was chosen because a combination of probing pocket depth and clinical attachment loss was the most common periodontitis case definition used in clinical studies.

### 2.8. Risk of Bias in Individual Studies

Assessment of risk of bias of individual studies was done using the Newcastle–Ottawa Quality Assessment Scale for cross-sectional [31], case-control [32], cohort studies [32].

## 3. Results

A total of 1982 records were identified through various databases and hand searching. After removing duplicates, 684 studies underwent title and abstract screening, and 622 were excluded. Of the 62 full-text articles assessed for eligibility, 47 were excluded for various reasons. Finally, 15 articles were included in the systematic review as seen in Figure 1 (based on PRISMA 2020 statement) [33].

### 3.1. Human Studies

This review included 15 observational studies from Brazil, Iran, Jordan, Egypt, India, and Turkey (Table 1). Among these, there were nine cross-sectional studies [34,35,36,37,38,39,40,41,42], two case-control [43,44], and four cohort studies [45,46,47,48]. The total study subjects from these studies were 6603 (males = 3432; females = 3171). Most studies showed a significant association between obesity and periodontitis [34,35,36,37,38,39,40,41,42]. Among these studies, a few showed obese females to be at a higher risk [35,37,45], and one study found no association between obesity and periodontal disease [44].

All the studies included used BMI to assess body composition. Most studies used the WHO classification to define obesity with a BMI cut-off of 29.9 or 30 kg/m^2^. Values such as BMI ≥ 25 kg/m^2^ [47] were also used to define obesity.

Periodontitis was defined using different criteria. Periodontitis was defined using CAL alone in three studies [35,45,48]. The remaining studies used CAL and PPD for the same [34,36,38,40,41,42,43,44,46]. In two studies, CAL was combined with CPI [31] and BOP [41]. The latest classification of periodontitis [49] was used in one study [48].

A few studies reported other periodontal parameters such as gingival inflammation [36,37], BOP [42], and tooth loss [41,46,47,48]. Tooth loss showed no association with increased BMI in a study [41].

Different sampling techniques, such as multistage probability sampling [35,45,46], systematic random sampling [36], convenience sampling [38,42], and consecutive sampling [41] have been used in these studies.

Some studies reported an increased risk of periodontitis in obesity with odds ratios (OR) of 2.1 [35], 1.77, 95% CI: 1.20–2.63 [46], 2.9, 95% CI: 1.3, 6.1 [36], 3.25, 95% CI: 1.27–8.31 [40], and 1.88, 95%CI: 1.05-3.37 [41]. A few other studies reported a relative risk (RR) of 1.64, 95% CI 1.11–2.43 [45], and 1.84 [47]. Positive correlations between BMI and PPD were reported by a few studies [37,43].

### 3.2. Animal Studies and Systematic Review 

Although the main aim of this review was to see if obesity increases the risk of periodontitis in humans, 12 animal studies were identified (Table 2) that evaluated this in different animal models such as C57BL/6 mice, Wistar rats, obese Zucker rats, and Holtzman rats. Various parameters including oral microbiota, inflammatory mediators, morphometric analysis of alveolar bone loss, gingival gene-expression pattern, body weight, proteome analysis, histopathological, histometric, and immunohistochemical, have been reported. Most studies showed an increased alveolar bone loss in periodontitis among obese animals [50,51,52,53,54,55,56]. Nonetheless, a few studies reported negligible or no effect of obesity on the progression of alveolar bone loss [57,58,59,60]. It was also found that hypothalamic obesity may produce a protective effect against periodontal disease [61].

Ten systematic reviews were identified on this topic (Table 3). Among these, most of the reviews and meta-analyses showed a positive association between obesity and periodontal disease. The total number of studies included in these reviews was 310 (ranging from 5–92 studies).

### 3.3. Risk of Bias across Studies

The risk of bias in individual observational studies assessed using the Newcastle–Ottawa Quality Assessment Scales showed a low risk of bias in seven studies, moderate risk in five studies, and high risk in three studies (Table 4).

Diet-induced obesity (DIO), tumor necrosis factor [TNF]-α), and bone metabolism (osteocalcin [O C], carboxy-terminal collagen crosslinks [CTX], and N-terminal propeptides of type I procollagen [P1NP]), tartrate-resistant acid phosphatase (TRAP), receptor activator of nuclear factor kappa beta ligand (RANKL), and osteoprotegerin (OPG), interleukin-6 (IL-6), cyclooxygenase-2 (COX-2),

## 4. Discussion

The overall findings of this systematic review show a link between obesity and periodontal disease in adult patients. Most of the studies reported a positive association between increased BMI and clinical attachment loss, probing pocket depth, or both [34,35,36,37,38,39,40,41,42,43,45,46,47,48]. Still, clarity is needed on the magnitude and mechanisms of this association. This review found more eligible cross-sectional studies and a few case-control and cohort studies that examined whether or not obesity is a risk factor or indicator for periodontal diseases. Many of these studies report higher odds of obese patients having periodontitis [35,36,37,41,43,45,46,47]. A dose-response relationship between obesity and periodontitis can be seen in some studies [35,36,42].

### 4.1. Mechanism Linking Obesity and Periodontal Disease

Obesity affects the immune responses due to an imbalance in the levels of pro-inflammatory cytokines in the plasma of obese subjects [17,69,70]. This increase might explain the relationship between obesity and periodontal disease. The large quantities of tumor necrosis factor-alpha (TNF-α), interleukin-6 (IL-6), interleukin-8 (IL-8), and plasminogen activator inhibitor-1 (PAI-1) secreted by adipose tissue are found to be proportional to the BMI [71] and the amount of adipose tissue present [19]. The adipokines cause low-grade inflammation due to gram-negative bacteria and inflammatory mediators [22,72]. The inflammatory and immune mediators in the body play a significant role in the pathogenesis and development of periodontitis. For example- IL-8 is a potent neutrophil chemoattractant that is increased in periodontitis patients. Similarly, TNF-α adversely affects the immunity of the body, especially in periodontal tissue that possibly acts as a risk factor for periodontal disease [9,73]. It can also worsen existing periodontitis disease by stimulating the fibroblasts and osteoclasts along with increased production of degrading enzymes [74]. Adipokine IL-6 destroys both periodontal alveolar bone and connective tissue [75,76]. Variations in leptin, adiponectin, and visfatin are also linked with obesity [13] and systemic inflammation [77]. Hence, there is an increased risk of periodontal inflammation and alveolar bone loss in obese patients [54,78].

### 4.2. Association between Obesity and Periodontal Disease-Human Observational Studies

Several epidemiological studies have reported a positive association between obesity and periodontal disease [34,35,36,37,38,39,40,41,42,43,45,46,47,48]. However, there was a contradicting result also [44]. Furthermore, it has been suggested that obesity is one of the strongest risk factors for inflammatory periodontal tissue destruction after smoking [79]. Several epidemiological studies supported the link between obesity and periodontal disease.

An association between obesity and periodontal disease was first reported in humans by Saito et al. [80]. Since then, many authors have studied the link between obesity and periodontal disease using various obesity parameters such as BMI, waist-hip ratio, body fat, etc. BMI has been used by most of the studies owing to its ease of use in all types of study settings and is the most commonly used parameter by WHO studies to indicate obesity [81].

Fat distribution plays a crucial role in the association with periodontitis [48,63,82]. High upper body obesity and high total body fat were correlated with a higher risk of periodontal disease [82]. Moreover, the pattern of fat distribution may impact the host immune response, causing glucose, lipid levels, and insulin resistance to increase [83].

Obesity is also related to higher gingival inflammation [36,37] and BOP [42]. No effect of general obesity on the percentage of BOP was observed independent of the analytical approach employed in another study [47]. This could be due to the partial mouth protocol for periodontal outcomes assessment, which could have misrepresented the prevalence or magnitude of the outcome [47].

A few studies suggest an association between obesity and tooth loss [41,46,47,48]. Obese patients were found to have fewer remaining teeth and higher levels of periodontitis, including deeper probing depths [46]. These patients were also more likely to be smokers and diabetics than those participants in the lower BMI categories [46]. There was an increase in the number of teeth lost as the BMI value increased [48]. However, some studies did not clarify if the teeth lost were due to periodontal disease [46,47,48], and in some cases, tooth loss was not associated with increased BMI [41].

Studies have indicated that a low physical activity and an unhealthy diet were significantly associated with increased odds of periodontal disease [42,72,84]. Subjects who exercised regularly were found to have lower plasma levels of pro-inflammatory markers and increased insulin sensitivity, which positively affects periodontal health [85,86,87,88].

A positive association between obesity and periodontitis among women has been noticed, and this association was more evident with the increase in obesity levels [35,37,45]. It is unclear why the increased tendency for periodontitis was not seen in males with higher BMI. Increased BMI in men could be due to muscle mass and the weight of bones [35]. Moreover, the pattern of body fat distribution in males is not similar to females. Therefore, increased BMI in males need not indicate obesity.

The study’s outcome may be influenced by the criteria used for defining obesity and periodontitis. Although all the studies included in this review used BMI to assess body composition, there were variations in the definition of obesity and its cut-off values. WHO classifies obesity based on BMI with a cut-off of 29.9 or 30 kg/m^2^. Some authors used BMI ≥ 25 kg/m^2^ [47] to define obesity. This could have led to variations in the observations of the studies. Similarly, there was no uniformity in the way periodontitis was defined. The original protocol included studies that considered CAL to define periodontitis. Since there were only a few studies, we opted to have studies that either utilized CAL alone or CAL and PPD together to identify periodontitis. CAL alone was used to define periodontitis in a few studies [35,45,48], although most studies used CAL and PPD for the same [34,36,38,39,40,41,42,43,44,46,89]. This has been the most common periodontitis case definition used in clinical studies [90,91]. Since we included studies from 2000 to 2022, we found that only one study [48] reported the use of the latest classification of periodontitis [49] in identifying periodontitis cases.

Appropriate sampling techniques are needed to reduce sampling errors and obtain a sample group that represents the population. Probability or random sampling techniques are preferred to obtain unbiased results and valid inferences [92]. Studies included in this review used different probability sampling techniques such as multistage probability sampling [35,45,46], and systematic random sampling [36]. It might be difficult to draw meaningful inferences from the studies that used non-probability sampling methods such as convenience sampling [33,37] and consecutive sampling [36] or from other studies that do not mention the sampling details.

### 4.3. Evidence from Animal Studies

Most of the animal studies showed an increased alveolar bone loss in periodontitis among obese animals [50,51,52,53,54,55,56]. The relationship between obesity and periodontal disease was first reported in obese Zucker rats. This animal model showed greater alveolar bone destruction in a ligature-induced periodontitis model [93]. Several other animal studies also reported evidence linking periodontal disease and obesity [50,57,59,61,93].

Perlstein and Bissada [93] showed both hyperplasia and hypertrophy of the walls of blood vessels supplying the periodontium in hypertensive and obese-hypertensive rats compared to non-obese controls. A study was conducted on diet-induced obese rats, the effect of obesity on innate immune responses to *Porphyromonas gingivalis* infection, an infection strongly associated with periodontitis [50]. The authors found a significantly higher level of alveolar bone loss in the obese rats than in the lean controls. However, another study on the effect of body weight on the pathogenesis of ligature-induced periodontal disease in Wistar rats failed to show the progression of alveolar bone loss [57]. Similarly, a few other studies also found no effect of obesity on the progression of alveolar bone loss [58,59,60].

Obesity-induced gingival oxidative stress positively correlated with a high serum-reactive oxygen metabolite in rats [59]. Brandelero et al. [61] found a decreased TNF-alpha gene expression in periodontal ligature-induced MSG-obese rats. They concluded that hypothalamic obesity might produce a protective effect against periodontal disease. Another study showed higher serum reactive oxygen species (ROS) levels in obese rats fed a high-fat diet without exercise [94], which led to the conclusion that obesity prevention by exercise training may effectively suppress gingival oxidative stress by decreasing serum ROS in rats.

### 4.4. Evidence from Systematic Review

Evidence from most of the reviews and meta-analyses shows a positive association between the role of obesity and periodontal disease. Although several unmeasured confounding factors could exist, a positive link between periodontal disease and obesity has been suggested despite variations in study populations [5,9,63,64,66]. Obesity and periodontitis have been correlated more strongly among women, non-smokers, and younger people than in the overall adult population [5]. Obese patients with periodontitis had more gingival inflammation than those with periodontitis patients who were not obese [67]. A positive association was also found between overweight/obesity and periodontitis during pregnancy [68]. Nonetheless, clarity is still needed on the direction, magnitude [10], and risk factors that exacerbate obesity and periodontitis [62]. Although inflammatory or metabolic parameters showed significant changes in overweight/obese patients than in normal-weight patients [65], this systematic review showed no difference in the clinical parameters.

However, some limitations must be considered. The design of these studies limits the interpretability of temporal relationships. Prospective studies are needed to substantiate the true nature of obesity as an independent risk factor for periodontitis.

### 4.5. Implications for Clinical Practice

Clinicians should stress the importance of maintaining a normal weight for healthy life. They should routinely record the body composition to assess the risk profile and bring awareness regarding the possible increased risk of periodontitis in their obese patients. Although an increased BMI could be a risk factor for periodontal diseases, practitioners should not only consider BMI values to assess overweight/obesity but also focus on other parameters for personalized management. Recommendations for future studies include recording visceral adiposity as it shows a more consistent association between obesity and periodontitis than BMI, which could add to risk stratification and raise patient awareness about their health condition.

## 5. Conclusions

This systematic review found a positive association between obesity in terms of increased BMI and periodontitis in adults. The studies demonstrated an association between an increased BMI and periodontal parameters such as CAL and pocket depth. A few studies also showed an increased risk of periodontitis in obese adults who were smokers. Based on the evidence obtained from this review, the BMI should be routinely assessed in patients to offer personalized management of periodontitis. Based on the findings of this review, we recommend the need to initiate awareness among clinicians and implement dental hygiene care prevention measures for obese patients.

## Figures and Tables

**Figure 1 ijerph-19-12684-f001:**
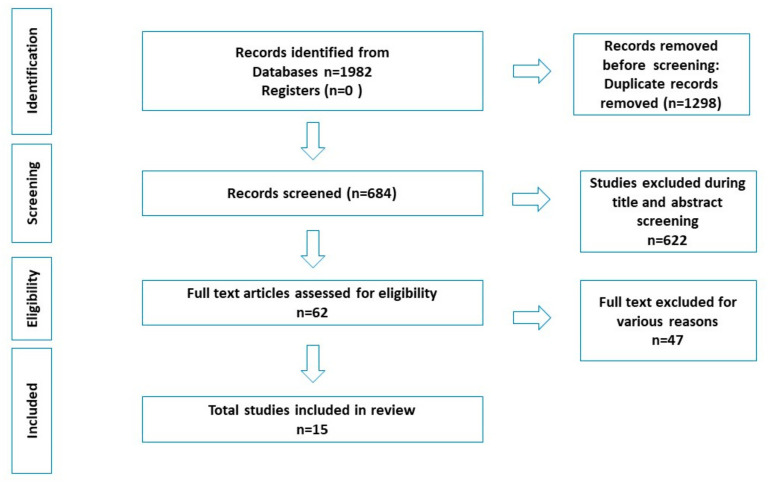
PRISMA Flowchart.

**Table 1 ijerph-19-12684-t001:** Human observational studies included in the systematic review.

Sl no	Author/Year Country/Study Design	Study Subjects	Age Range (Mean) Male/Female	Body Composition Criteria (BMI)	Definition of Periodontitis Used	Secondary Parameters (Tooth Loss, BOP)	Sampling Method	Inclusion of Smokers/Diabetics	Main Observations
1	Dalla et al. 2005 [35] Brazil Cross-sectional study	706 individuals (131 normal/ 134 overweight/ 64 obese	30–65 years 329 males 377 females	* Four BMI categories (WHO criteria)	Individuals with ≥30% teeth with attachment loss ≥5 mm	Not mentioned	Multistage probability sampling method	Smokers and Diabetes (self-reported)	Higher risk of periodontitis among obese females than normal females (OR = 2.1). There is no significant association between overweight and periodontitis among females and the age group above 35 years—more pronounced association of BMI and periodontitis in non-smokers.
2	Sarlati et al. 2008 [43] Iran Case-control study	80 young volunteers (40 normal/40 overweight/obese)	18–34 years; obese individuals (29.1 + 4.7 years) and normal individuals (24 + 5 years) 10 males 70 females	* Four BMI categories (WHO criteria)	PPD and CAL	Not mentioned	Not mentioned	Smokers and Diabetes (self-reported)	Positive correlations between BMI and PPD (R = 0.33) BMI and CAL (R = 0.39). Not adjusted for confounding.
3	Khader et al. 2009 [36] Jordan Cross-sectional study	340 persons (13 underweight/108 normal/115 overweight/104 obese)	18–70 years 168 males 172 females	* Two BMI categories (WHO criteria) obese and overweight	Four or more teeth with one or more sites with PPD ≥4 mm, CAL ≥ 3 mm.	Obese patients had a significantly higher average of GI.	Systematic random sampling	Smokers and Diabetes (self-reported)	Higher risk of periodontitis in obese patients (OR = 2.9; CI 1.3, 6.1). Insignificant association between smoking and the prevalence of periodontal disease.
4	Amin 2010 [37] Egypt Cross-sectional study	380 adults (92 normal/161 overweight/127 obese)	20–26 years 170 males 210 females	* 3 BMI categories (WHO criteria) normal weight, overweight, obese	CAL, GI, and CPI	Significant correlation between BMI and G.I.	Not mentioned	No	High correlation between CAL and BMI (r = 0.9, *p* < 0.01) in young females than in males.
5	Pataro et al. 2012 [42] Brazil Cross-sectional study	594 females (352 normal/54 overweight/48 obesity level I/56 obesity level II/74 obesity level III)	18–65 years; 39.7 ± 17.35 years 594 females	Normal weight (BMI 20–24.99 kg/m^2^) overweight (BMI 25–29.99 kg/m^2^), ** obesity level I, obesity level II and obesity level III (WHO 1997)	Proximal CAL ≥ 4 mmin two or more teeth, or proximal PD ≥ 5 mm in two or more teeth (Page et al. 2007)	BOP was more prevalent in the obese group III (34.8%, *p* < 0.001)	Convenience sampling	Both were included but unclear how it was evaluated.	Statistically significant differences in BOP, PPD, CAL ≥ 4 mm (*p* < 0.05) among obese and overweight women as compared to women with normal BMI. BMI > 30 kg/m^2^ interacted with diabetes (4.03), and smoking (15.79) (*p* < 0.03). The association was more evident as obesity increased.
6	Budduneli et al. 2014 [44] Turkey Case-control study	91 females (31 normal and 60 obese)	43.10 ± 10.87 years 91 females only	Obesity is diagnosed based on the WHO criteria (not specified)	Not clearly defined. PPD, CAL, and dichotomous BOP (present or absent within 10 s after probing) recorded	Not mentioned	Not mentioned	Smokers (self-reported). Diabetes excluded.	BMI did not correlate to clinical periodontal parameters in the obese group (but correlated with serum levels of inflammatory molecules (*p* < 0.05). Smokers (obese and non-obese) did not show significant differences in P.D., BOP, and PI (*p* > 0.05). Obese non-smokers had higher periodontitis CAL, BOP and PI (*p* < 0.05). PD was the same in obese and control groups of non-smokers.
7	Gaio et al. 2016 [45] Brazil Cohort study	583 individuals (297 normal/177 overweight/108 obese)	36.02 ± 14.97 years 333 males 249 females	* Four BMI categories (WHO criteria)	Proximal PAL ≥ 3 mm in ≥ 4 teeth over the 5 years of follow-up.	Not mentioned	Multistage probability sampling strategy	Smokers (self-reported) Diabetes excluded.	Higher risk of PAL in obese females than normal weight females (R.R. = 1.64, 95% CI = 1.11–2.43) and males. No statistically significant associations were observed between obesity and PAL progression for never or ever smokers.
8	Deshpande and Amrutiya 2017 [38] India Cross-sectional study	100 patients with chronic generalized periodontitis/gingivitis (50 normal/50 obese)	18–63 years Mean age 34.14 ± 11.70 (non-obese) and 34.02 ± 9.03 (obese) 63 males 37 females	Obese (BMI > 30) Non-obese (BMI < 30)	PPD and CAL	Not mentioned	Convenience sampling	Unclear	Higher prevalence of periodontitis in obese patients than in the control group (*p* < 0.05 for PPD, and *p* < 0.031 for CAL).
9	Nascimento et al. 2017 [47] Brazil Cohort study	1076 individuals	20–59 years 463 males 603 females	Obese (BMI ≥ 25 Kg/m^2^)	Combination of CAL and BOP	Tooth loss was mentioned but not mentioned if due to periodontal disease	Not mentioned	Smokers and diabetes (self-reported)	A higher risk of attachment loss and BOP in obese patients presented (RR 1.45 for AL and BOP in different teeth; RR 1.84 for AL and BOP in the same tooth).
10	Santos et al. 2019 [40] Brazil Cross-sectional study	236 individuals (156 normal/ 69 overweight 80 obese	18–34; 35 and above 52 males 184 females	* Two BMI categories (WHO criteria) overweight, obese.	Based on CDC-AAP case classification	Not mentioned	Not mentioned	Smokers (self-reported) Diabetes excluded.	Positive association between severe periodontitis and obesity (OR = 3.25, 95% CI = 1.27–8.31, *p* = 0.01) but not with overweight (*p* = 0.59).
11	Gulati et al. 2020 [39] India Cross-sectional study	317 individuals (52 overweight/ 251 obesity I/ 9 obesity II 5 obesity III)	25–70 years 203 males 114 females	** Obese Class I, Class II, Class III	Four or more teeth with one site or more with PPD ≥ 4 mm and CAL ≥ 3 mm was present.	Not mentioned	Not mentioned	Unclear	Deeper PD was significantly associated with obesity determinants, especially among Class 2 and Class 3 obese individuals with chronic periodontitis.
12	Maulani et al. 2021 [41] Indonesia Cross-sectional study	262 individuals (135 normal/ 127 overweight or obese)	18–66 years 105 males 157 females	* Four BMI categories (WHO criteria by the Asia-Pacific perspective)	CAL 5 mm and PD 6 mm were cut-off measurements between mild and severe periodontitis	Yes; not associated with increased BMI	Consecutive sampling	Yes, but unclear how it was recorded	Increased BMI showed a positive correlation with periodontitis of all severity. (aOR = 1.88, 95%CI 1.05-3.37; *p* < 0.05). Lower BMI is found in smokers than in non-smokers.
13	Carneiro et al. 2022 [34] Brazil Cross-sectional study	345 individuals (133 normal/ 106 obese)	49.08 years (±) 14.26 92 males 253 females	* Six BMI categories (WHO criteria) low weight, normal weight, overweight, ** obese I, obese II, obese III	CDC/AAP criteria	Not mentioned	Not mentioned	Smokers (self-reported) Unclear how diabetes was recorded.	Females and younger participants showed a positive association between obesity and periodontitis.
14	Cetin et al. 2022 [48] Turkey Retrospective study	142 with periodontitis (59 normal/62 overweight/21 obese)	above 18; 57.24 ± 8.78 82 males 60 females	* Three BMI categories (WHO criteria) normal weight, overweight, obese	interdental CAL at the site of greatest loss (staging and grading)	number of remaining teeth	not mentioned	Smokers (self-reported) Diabetic status obtained from the ‘patient’s hospital records.	CAL (*p* < 0.001), PPD (*p* < 0.05), PI (*p* < 0.05)), stage and grade of periodontitis (*p* < 0.05) were higher overweight and obese patients. BMI and smoking status showed no significant association (*p* = 0.142). Overweight and obese patients were at higher risk of developing stage III–IV periodontitis
15	Linden et al. 2007 [46] UK. Retrospective cohort study	1362 males (336 normal/ 728 overweight/ 298 obese)	60–70 years; 64 ± 2.9 1362 males	Four BMI categories (WHO criteria)	High-threshold periodontitis was identified when ≥ 15% of all sites measured had a loss of attachment ≥6 mm, and there was at least one site with deep pocketing (≥6 mm).	Tooth loss mentioned	the multistage probability sampling method.	Smokers and Diabetes (self-reported)	Strong association between BMI and high-threshold periodontitis for heavy smokers (OR 4.21, 95% CI% 2.04–8.72, *p* 0.0001) and light smokers (OR 3.22, 95% CI% 1.76–5.88, *p* 0.0001) among older men. High BMI levels in early life did not predict periodontitis in later life in the men studied.

Clinical attachment loss (CAL); gingival index (GI); Community Periodontal Index (CPI); bleeding on probing (BOP); periodontal attachment loss (PAL); Centers for Disease Control and Prevention-American Academy of Periodontology (CDC-AAP), * BMI categories (WHO criteria)- underweight (BMI < 18.5 kg/m^2^), normal weight (18.5 to 24.9 kg/m^2^), overweight (25 to 29.9 kg/m^2^) and obese (≥30 kg/m^2^); ** obesity level I (BMI 30–34.99 kg/m^2^), obesity level II (BMI 35–39.99 kg/m^2^), obesity level III (BMI ≥ 40 kg/m^2^); odds ratio (OR); adjusted odds ratio (aOR).

**Table 2 ijerph-19-12684-t002:** Animal studies included in the review.

Sl No	Author and Year Country	Study Subjects	Parameters Studies	Major Observations
1	Amar et al. 2007 [50] USA	DIO mice and lean control C57BL/6 mice were infected orally	Oral microbial sampling, inflammatory response, (TNF-alpha, IL-6, and serum amyloid A (SAA).	Obesity causes immune dysregulation. It also interferes with the ability of the immune system to respond to *P. gingivalis* infection. Increased alveolar bone loss after bacterial infection was observed in mice with DIO.
2	Simch et al. 2008 [57] Brazil	30 female Wistar rat. Test group (n = 14 rats on cafeteria diet) control group (n = 16 on regular).	Morphometric analysis of alveolar bone loss by standardized digital photographs (software Image Tool 3.0).	No statistically significant differences between alveolar bone loss of test animals and controls. Progression of alveolar bone loss in rats not influenced by obesity.
3	Tomofuji et al. 2009 [59] Japan	28 rats. The obese Zucker rats (n = 14) lean littermates (n = 14)	8-hydroxydeoxyguanosine, ratio of reduced/oxidized glutathione, serum level of reactive oxygen metabolites, gingival gene-expression pattern.	Obese rats had higher levels of gingival 8-hydroxydeoxyguanosine. There was also a decreased ratio of reduced/oxidized glutathione with increasing serum reactive oxygen metabolites. No significant differences in the degree of alveolar bone loss between lean and obese rats Gene expressions related to a capacity for xenobiotic detoxification were downregulated in obese rats.
4	Verzeletti et al. 2012 [58] Brazil	24 female Wistar rats cafeteria diet (n = 13) regular diet (n = 11)	Body weight, Morphometric registration of alveolar bone loss.	Alveolar bone loss was not statistically different between obese and non-obese group
5	Brandelero et al. 2012 [61] Brazil	20 newborn male Wistar rats MSG group (n = 10) Control group (n-10)	Radiographic analyses of alveolar bone resorption, Tumor Necrosis Factor α (TNFα), Gene expression in gingival tissue.	The alveolar bone resorption was 44% lower in MSG-obese rats compared with control rats. Hypothalamic obesity may produce a protective effect against periodontal disease
6	Cavagni et al. 2013 [52] Brazil	28 Wistar rats. Control group (n = 10) Test group (cafeteria diet: n = 10)	Morphometric analysis of standard digital photographs, Mean alveolar bone loss.	Animals in the test group showed 20 sites with spontaneous periodontal disease, whereas in control animals, only 8 sites exhibited periodontal breakdown. Obesity increases the occurrence of spontaneous periodontal disease in Wistar rats.
7	Cavagni et al. 2016 [51] Brazil	60 male Wistar rats. Control group (n = 15) periodontitis (n = 15) obesity/hyperlipidemia (n = 15) obesity/hyperlipidemia plus periodontitis (n = 15).	Body weight and Lee index, Serum glucose and cholesterol/ triglycerides, alveolar bone loss (micro CT), Serum tumor necrosis factor (TNF)-α, Interleukin (IL)-1β.	Groups exposed to CAF exhibited higher ABL in the sides without ligature. No differences were observed among groups for IL-1β and TNF-α. Obesity and hyperlipidemia modulate the host response to challenges in the periodontium, increasing the expression of periodontal breakdown.
8	Muluke et al. 2016 [53] USA	Four-week-old male C57BL/6 mice (n = 10 per group) high-fat diet (HFD) normal caloric diet	percentage fat, serum inflammation (TNF-α, OC, CTX, P1NP markers	Alveolar bone loss was significantly greater in obese animals. Osteoclasts also showed an augmented inflammatory response to *P. gingivalis* in obese animals. High-fat diet was more important than obesity in affecting alveolar bone loss.
9	Zuza et al. 2018 [54] Brazil	48 adult Wistar rats High fat diet group (n = 24) Normal diet group (n = 24)	Histopathological, histometric, and immunohistochemical analyses. TRAP, RANKL, OPG via immunolabeling.	Histology shows that inflammation lasted longer in obese rats. Obesity induced by a high-fat diet caused more severe local inflammatory response and alveolar bone loss.
10	Damanaki et al. 2018 [55] Germany	12 C57BL/6 mice Younger lean mice (n = 4) Older lean mice (n = 4) Younger obese mice (n = 4)	IL-6, COX-2, visfatin and adiponectin in gingival samples (real-time PCR)	Alveolar bone loss was significantly lower in the older mice as compared to the younger animals. Gingival COX-2 and visfatin expressions were higher in the obese versus lean mice and in the older versus younger mice
11	Damanaki et al. 2021 [56] Germany	15 Wistar rats High-fat diet (n = 15) Normal diet (n = 15)	Histomorphometry to assess healing, TRAP staining and immuno-histochemistry for RUNX2 and osteopontin.	Spontaneous bone healing in periodontal defects is affected by obesity even in the presence of regeneration-promoting molecules like EMD.
12	Lopes et al. 2022 [60] Brazil	16 Holtzman rats were ligature-induced periodontitis (n = 8). Obesity plus ligature-induced periodontitis (O.P.) (n = 8)	Body weight, adipose tissue weight, and blood test, Bone loss (micro-CT and histologic analyses), Proteome analysis from the periodontal ligament tissues (PDL), Immunohistochemistry for spondin1, vinculin, and TRAP.	Histologically, it was found that obesity did not significantly affect bone loss resulting from periodontitis. Obesity affects the proteome of PDL submitted to experimental periodontitis.

**Table 3 ijerph-19-12684-t003:** Systematic reviews and meta-analysis included in the review.

Sl. No	Author/Year/Country	Number of Studies	Study Period	Major Observations
1	Chaffee and Weston 2010 [5] USA	57	Up to 2010	In total, 41 studies suggested a positive association consistent with a biologically plausible role for obesity in the development of periodontal disease. The fixed-effects summary odds ratio was 1.35, with some evidence of a stronger association among younger adults, women, and non-smokers. Also, a greater mean CAL among obese individuals, a higher mean BMI among periodontal patients, and a trend of increasing odds of prevalent periodontal disease with increasing BMI.
2	Suvan et al. 2011 [10] U.K.	33	Up to 2009	There were 19 studies included in the meta-analyses. Statistically significant associations between periodontitis and BMI category obese OR 1.81(1.42, 2.30), overweight OR 1.27(1.06, 1.51), and obese and overweight combined OR 2.13(1.40, 3.26). Support an association between BMI overweight and obesity and periodontitis, although the magnitude is unclear.
3	Moura-Grec et al. 2014 [62] Brazil	31	Up to 2010	A positive association in 25 studies (not associated in 6 studies). The meta-analysis showed a significant association with obesity and periodontitis (OR = 1.30 [95% Confidence Interval (CI), 1.25–1.35]) and with mean BMI and periodontal disease (mean difference = 2.75). Obesity was associated with periodontitis, however, the risk factors that aggravate these diseases should be better clarified to elucidate the direction of this association.
4	Keller et al. 2015 [63] Denmark	13	Up to June 2014	Two longitudinal studies found a direct association between being overweight and the subsequent risk of developing periodontitis. Three studies found a direct association between obesity and the development of periodontitis among adults. Two intervention studies on the influence of obesity on periodontal treatment effects found that the response to non-surgical periodontal treatment was better among lean than obese patients. The remaining three studies did not report treatment differences between obese and lean participants. Among the eight longitudinal studies, one study adjusted for C-reactive protein (CRP), and biologic markers of inflammation such as CRP, interleukin-6, and tumor necrosis factor-α, and inflammation markers were analyzed separately in three of the five intervention studies. This systematic review suggests that overweight, obesity, weight gain, and increased waist circumference may be risk factors for the development of periodontitis or worsening of periodontal measures.
5	Nascimento et al. 2015 [64] Brazil	5	Up to Feb 2015	Subjects who became overweight and obese presented a higher risk of developing new cases of periodontitis (RR 1.13; 95%CI 1.06–1.20 and RR 1.33 95%CI 1.21–1.47 respectively) compared with counterparts who stayed at a normal weight. A clear positive association between weight gain and new cases of periodontitis was found. However, these results originated from limited evidence. Thus, more studies with prospective longitudinal designs are needed.
6	Papageorgiou et al. 2015 [65] Germany	15	Up to July 2013	No difference was found in clinical periodontal parameters, but significant differences in inflammatory or metabolic parameters were found between overweight/obese and normal-weight patients. Existing evidence is weak.
7	Martinez-Herrera et al. 2017 [9] Spain	28	2000-2017	A total of 26 studies described an association between obesity and periodontal disease (no association n = 2). The development of insulin resistance as a consequence of a chronic inflammatory state and oxidative stress could be implicated in the association between obesity and periodontitis.
8	Khan et al. 2018 [66] Australia	25	2003 and 2016	There were 25 eligible studies from 12 countries. 17 showed an association between obesity and periodontitis (odds ratios ranged from 1.1 to 4.5). The obesity indicators of BMI, waist circumference, waist-hip ratio, and body fat percentage were significantly associated with measures of periodontitis of bleeding on probing, plaque index, probing depths, clinical attachment loss, calculus, oral hygiene index, and community periodontal index. Two prospective cohort studies in the review showed no significant association between obesity and periodontitis, but these studies had limitations in study design and used inappropriate epidemiological diagnostic measures of periodontitis. Evidence suggests that obesity is associated with periodontitis in adolescents and young adults.
9	da Silva et al. 2021 [67] Brazil	92	upto Jan 2021	Ninety studies were included (cross-sectional/clinical trials [n = 82], case-control [n = 3], cohorts [n = 5]). Most of the studies demonstrated no significant difference in the measures of gingival inflammation regardless of the comparison performed. Meta-analysis showed that among individuals with periodontitis, significantly higher levels of gingival inflammation are observed in those with obesity (n of individuals = 240) when compared to those who were not obese (n of individuals = 574) (SMD:0.26; 95%CI:0.07–0.44). When considering population-based/those studies that did not provide periodontal diagnosis, significantly higher measures of gingival inflammation were observed in the groups with higher BMI.
10	Foratori-Junior et al. 2022 [68] Brazil	11	2000–2021	11 studies were included. Most studies had a low risk of bias. A positive association between overweight/obesity and periodontitis was found, with an average of 61.04% of women with overweight/obesity and periodontitis, showing the overall random-effects relative risk and 95% CI of 2.21 (1.53–3.17) (*p* < 0.001). A positive association was found between overweight/obesity and periodontitis during pregnancy.

**Table 4 ijerph-19-12684-t004:** (**a**): Risk of bias in individual cross-sectional studies using The Newcastle–Ottawa Quality Assessment Scale by Dubey et al. 2022 [31]. (**b**): Risk of bias in individual case-control and cohort studies using the Newcastle–Ottawa Quality Assessment Scale by Stang et al. 2010 [32].

(a)										
Cross-Sectional Studies										
Sl. No	Author; Year	Selection				Comparability	Outcome			Total (Out of 10)
		Representativeness of the sample	Sample size	Non-respondents	Ascertainment of the exposure (risk factor):	Comparability of different outcome groups based on the design or analysis	Ascertainment of outcome	The same method of ascertainment for cases and controls		
1	Dalla et al. 2005 [35]	1	1	1	2	2	1	1		9 (low bias)
2	Khader et al. 2009 [36]	1	1	1	2	2	1	1		9 (low bias)
3	Amin 2010 [37]	1	1	0	2	0	1	0		5 (high bias)
4	Pataro et 2012 [42]	1	1	1	1	1	1	1		7 (moderate bias)
5	Deshpande and Amrutiya 2017 [38]	1	1	1	1	0	1	1		6 (high bias)
6	Santos et al. 2019 [40]	1	1	1	1	1	1	1		7 (moderate bias)
7	Gulati et al. 2020 [39]	0	0	1	1	1	1	1		5 (high bias)
8	Maulani et al. 2021 [41]	1	1	1	1	1	1	1		7 (moderate bias)
9	Carneiro et al. 2022 [34]	1	1	1	1	1	1	1		7 (moderate bias)
**(b)**										
**Case Control Studies**										
**Sl. No**	**Author; Year**	**Selection**				**Comparability**	**Exposure**			**Total** **(out of 9)**
		Adequate definition	Representativeness of case	Selection of Control	Definition of control	Comparability of cases and controls based on the design or analysis	Ascertainment of exposure	The same method of ascertainment for cases and controls	Non-response rate	
1	Sarlati et al. 2008 [43]	1	1	1	0	1	1	1	0	6 (moderate bias)
2	Budduneli et al. 2014 [44]	1	1	1	1	1	1	1	1	8 (low bias)
**Cohort Studies**										
**Sl. No**	**Author; Year**	**Selection**				**Comparability**	**Outcome**			**Total** **(out of 9)**
		Representativeness of the exposed cohort	Selection of the non-exposed cohort	Ascertainment of exposure	Shows that outcome of interest was not present at the start of the study	Comparability of cohorts based on the design or analysis	Assessment of outcome	Was follow-up long enough for outcomes to occur	Adequacy of follow-up of cohorts	
1	Liden et al. 2007 [46]	1	1	1	1	2	1	1	1	9 (low bias)
2	Gaio et al. 2016 [45]	1	1	1	1	1	1	1	1	8 (low bias)
3	Nascimento et al. 2017 [47]	1	1	1	1	1	1	1	1	8 (low bias)
4	Cetin et al. 2022 [48]	1	1	1	1	1	1	0	1	7 (low bias)

## Data Availability

Not applicable.

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
