# Peer review of "Is Obesity a Risk Factor for Periodontal Disease in Adults? A Systematic Review"

_ijerph, 2022, doi:10.3390/ijerph191912684_

Round 1

Reviewer 1 Report

Dear Authors,

This systematic review aims to explore the possibility of significant evidence on the association between obesity and periodontitis and to determine the necessity to consider obesity as a risk factor for periodontitis.

The study was very interesting and it is of scientific interest and in line with the aims of the journal. 

However, there are some issues that should be addressed. 

Introduction

-       In my opinion you should add a brief section in which is reported the link among periodontal and systemic diseases. Pleass discuss and cite: “Periodontal Disease and Vitamin D Deficiency in Pregnant Women: Which Correlation with Preterm and Low-Weight Birth? J Clin Med. 2021 Oct 2;10(19):4578. doi: 10.3390/jcm10194578.” ; “Periodontitis and cardiovascular diseases: Consensus report. J Clin Periodontol. 2020 Mar;47(3):268-288. doi: 10.1111/jcpe.13189.” ; “Periodontal disease and cancer: Epidemiologic studies and possible mechanisms. Periodontol 2000. 2020 Jun;83(1):213-233. doi: 10.1111/prd.12329.” ; “Functional status and oral health in patients with amyotrophic lateral sclerosis: A cross-sectional study. NeuroRehabilitation. 2021;48(1):49-57. doi: 10.3233/NRE-201537.”

Materials and Methods

-       Figure 1: Please use the 2020 Prisma Flow Diagram (Page MJ, McKenzie JE, Bossuyt PM, Boutron I, Hoffmann TC, Mulrow CD, et al. The PRISMA 2020 statement: an updated guideline for reporting systematic reviews. BMJ 2021;372:n71. doi: 10.1136/bmj.n71. For more information, visit: http://www.prisma-statement.org/).

Discussion

This section was well written.

Author Response

Dear Reviewer,

Thank you for taking the time to review our paper. Please find the point-by-point response attached here. 

Reviewer 2 Report

Dear authors,

Thank you for submitting the current manuscript.

I hope that my suggestions will help you increase the quality of it.

This study investigates: Obesity as a Risk Factor for Periodontal Disease in Adults.

The abstract is organized properly, briefly describing the systematic review research.

Introduction

This part decribes clearly the problem of obesity and periodontium inflammation process.

Materials and Methods

The aim of the research and research questions in this part are not presented properly. In my opinion it would be better to number them from 1-3.

Wrong numbering of the parts of the Materials and Methods. There is 1.1. for each one. Should be changed.

Results

Wrong numbering of the parts of the research, Introduction, Materials and Methods, Results,

Conclusions

This part is made properly, with most important issues.

The article is made very precisely and great effort was made in making this research.

Moderate English corrections are necessary.

Best regards

Author Response

(The authors gave the same response as above.)
